# Effects of oxygen-prebreathing on tissue nitrogenation in normobaric and hyperbaric conditions

**Edward Tom Ashworth**[1], **Ryotaro Ogawa**[2], **David Robert Vera**[2], **Peter Lindholm**[1,2]*

**1** Department of Emergency Medicine, University of California San Diego, La Jolla, CA, United States of America, **2** Department of Radiology, University of California San Diego, La Jolla, CA, United States of America

* plindholm@health.ucsd.edu

**Data Availability Statement:** All relevant data are within the manuscript.

**Funding:** Funding was obtained by PL from the Office of Naval Research (N00014-20-1-2763;

## Abstract

### Background

Breathing pure oxygen causes nitrogen washout from tissues, a method commonly deployed to prevent decompression sickness from hypobaric exposure. Theoretically pre-breathing oxygen increases the capacity for nitrogen uptake and potentially limits supersaturation during dives of short duration. We aimed to use $^{13}N_2$, a radioactive nitrogen isotope, to quantify tissue nitrogen following normobaric and hyperbaric exposures.

### Methods

Twenty Sprague Dawley rats were divided in 4 conditions; normobaric prebreathe, normobaric control, hyperbaric prebreathe, hyperbaric control. Prebreathed rats breathed oxygen for 1 h prior to the experiment whilst controls breathed air. Normobaric rats breathed air containing $^{13}N_2$ at 100 kPa for 30 min, whereas hyperbaric rats breathed $^{13}N_2$ at 700 kPa before being decompressed and sedated using air-isoflurane (without $^{13}N_2$ for a few minutes). After euthanization, blood, brain, liver, femur and thigh muscle were analyzed by gamma counting.

### Results

At normobaria prebreathing oxygen resulted in higher absolute nitrogen counts in blood (p = .034), as well as higher normalized counts in both the liver and muscle (p = .034). However, following hyperbaric exposure no differences were observed between conditions for any organ (p>.344). Both bone and muscle showed higher normalized counts after hyperbaria compared to normobaria.

### Conclusions

Oxygen prebreathing caused nitrogen elimination in normobaria that led to a larger "sink" and uptake of $^{13}N_2$. The lack of difference between conditions in hyperbaria could be due to the duration and depth of the dive mitigating the effect of prebreathing. In the hyperbaric

https://www.nre.navy.mil/). The funders had no role in study design, data collection and analysis, decision to publish, or preparation of the manuscript.

**Competing interests:** The authors have declared that no competing interests exist.

conditions the lower counts were likely due to off-gassing of nitrogen during the sedation procedure, suggest a few minutes was enough to off-gas in rodents. The higher normalized counts under hyperbaria in bone and muscle likely relate to these tissues being slower to on and off-gas nitrogen. Future experiments could include shorter dives and euthanization while breathing $^{13}N_2$ to prevent off-gassing.

## Introduction

One of the main risks of diving is decompression sickness (DCS; [1]). Among those at risk are those for who this is an occupational exposure, namely scientific and naval divers, as well as civilian divers. Additionally, aviators and astronauts are at risk of DCS, although this is as a result of rapidly ascending to altitude, or departing a pressurized space habitat, rather than ascending from depth. DCS is characterized by the formation of bubbles that appear during ascent when the partial pressure of dissolved nitrogen exceeds the ambient pressure. Bubble formation occurs in health, with bubbles being filtered out of the circulation by the lungs without reaching the left side of the heart [2]. However, when ascent is too rapid these bubbles can form within tissues leading to spinal cord injury and musculoskeletal pain.

Oxygen prebreathing (OPB) is currently used by aviators [3], during spacewalks [4], and during emergency submarine rescue [5] to denitrogenate tissues and thereby minimize the risk of DCS. During OPB nitrogen is expired from venous blood, creating a gradient for tissue nitrogen to move into venous blood and be expired. By lowering tissue nitrogen content, the amount of bubbles forming during decompression declines, thereby minimizing the risk of injury. However, how each organ is affected is unknown and frequently based on retrospective analysis. Current operational procedures are based on empirical evidence that enables retrofitted mathematical modelling to approximate risks [6]. These models use simulated tissue compartments with theoretical half-lives that correspond to observed DCS cases in humans and animals, which may not reflect real tissue nitrogen kinetics.

We have recently built on the method of Iwata et al. [7] to use nitrogen-13 ($^{13}N_2$), a radio-isotope of nitrogen, to track nitrogen movement through the body [8]. We aim to use this method to determine the effectiveness of OPB on minimizing DCS risk at both normobaria and hyperbaria. Firstly, we aim to determine whether $^{13}N_2$ can be used to assess organ-specific nitrogen loading at both normobaric and hyperbaric conditions. Within these experiments we hypothesize that the OPB will washout nitrogen from tissue, resulting in them accruing more $^{13}N_2$ during the experimental period.

## Materials and methods

Twenty Sprague Dawley rats were used in accordance with the Institutional Animal Care and Use Committee (UCSD IACUC–S19054) ethical approval and guidelines, including all euthanasia processes. All experimental staff were trained in animal handling in accordance with UCSD IACUC guidelines. Rats were obtained from the same litter (Charles River Laboratories, Wilmington, MA) and were housed in pairs with a standard diet. Ten rats were assigned to the normobaria condition, with the remaining ten assigned to hyperbaria (Fig 1). During the experiments all rats breathed nitrogen-13 laden air. All rats were weighed (CP153, Sartorius, Germany) prior to experiments.

Nitrogen-13 gas ($^{13}N_2$) was created off-site (PetNet, Siemens Healthineers, CA, USA) and delivered to the lab, housed within a vial of ammonium. The $^{13}N_2$ vial was placed in a dose

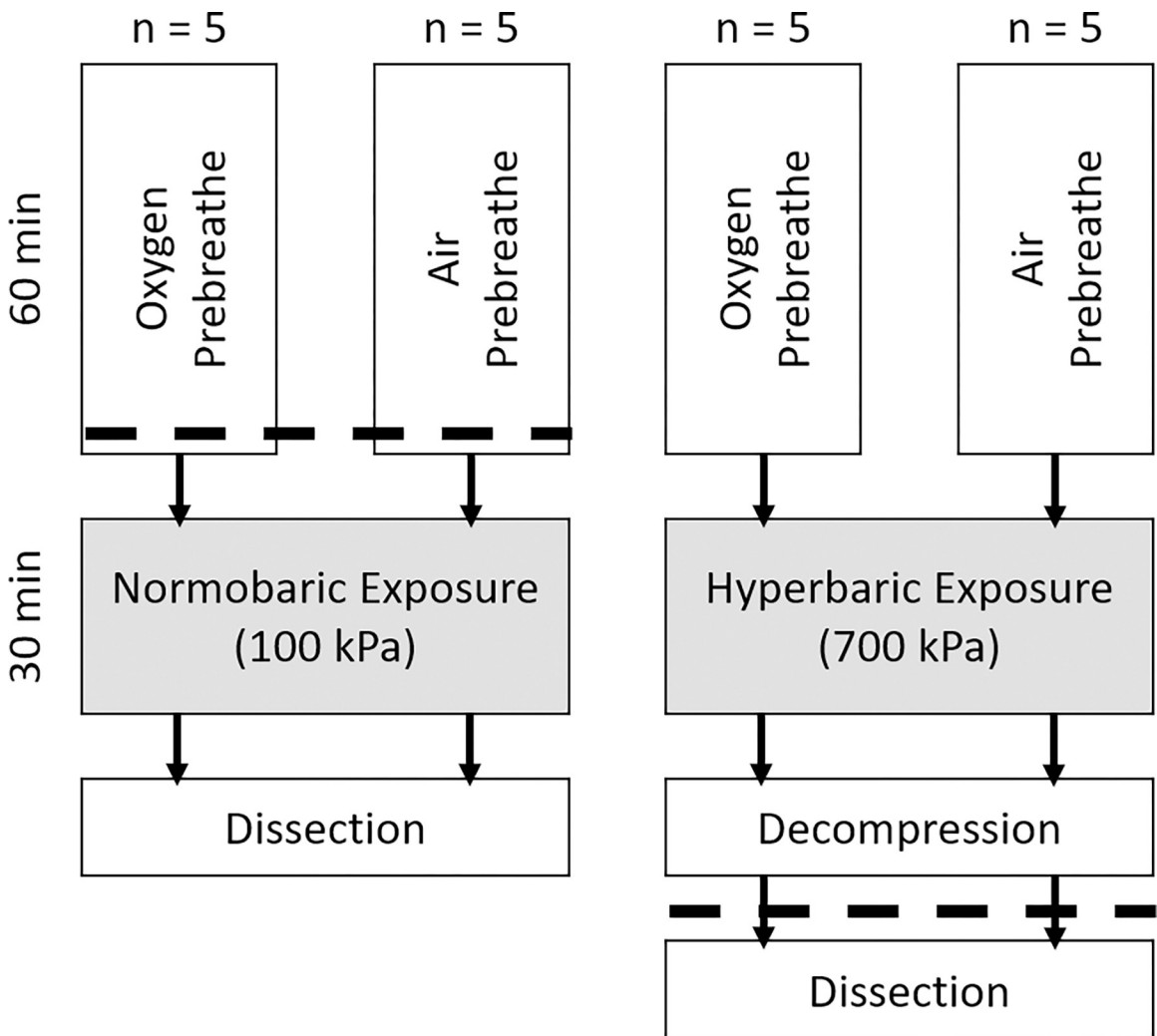

**Fig 1. Schematic showing the time course of the experiments.** The dashed line indicated the onset of anesthesia using isoflurane.

calibrator (CRC-15W, Capintec, NJ, USA) to obtain baseline activity. The $^{13}N_2$ solution was bubbled using air, at a rate of 1 L.min$^{-1}$ for 3 min, to release the $^{13}N_2$ [8]. During normobaric experiments the $^{13}N_2$ was bubbled into a non-diffusing bag connected to a ventilator (Physio-Suite, Kent Scientific, CT, USA), while during hyperbaric experiments the $^{13}N_2$ was bubbled directly into the closed hyperbaric chamber. The $^{13}N_2$ vial was then placed back into the dose calibrator to determine the amount of activity that was extracted. As $^{13}N_2$ has a 10 min half-life, all experiments were designed to be completed within 100 min of the $^{13}N_2$ generation to ensure sufficient radioactivity readings [8].

## Normobaric experiments

Prior to experimentation the rat was randomized using a balanced random number generator to receive either air or OPB, which was done for 60 min (Fig 1). Rats were anesthetized using isoflurane (3%) and intubated approximately 15 min before arrival of $^{13}N_2$ (Fig 2) and connected to the ventilator (90 breaths per min, tidal volume = 1% of body mass), all whilst still breathing normal air. Once the $^{13}N_2$ was in the non-diffusing bag the ventilator inlet was switched to the $^{13}N_2$ bag to begin the experiment (Fig 2). The rat remained anesthetized

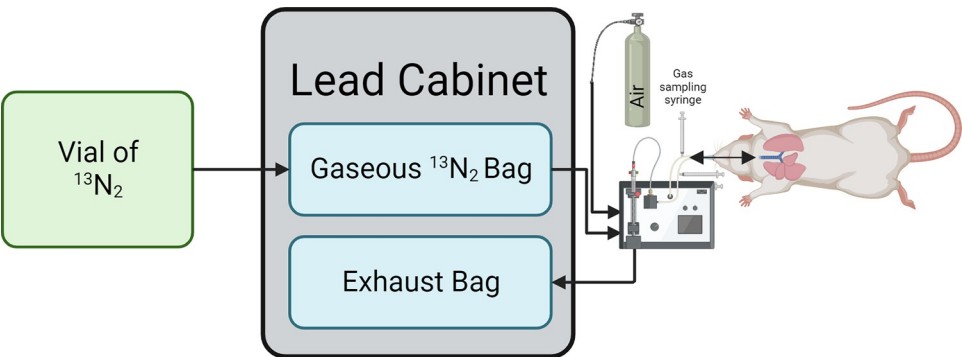

**Fig 2. Normobaric experimental setup.** The $^{13}N_2$ is bubbled out of liquid into a non-diffusing bag housed within a lead cabinet. The ventilator draws from this bag to ventilate the rat, while both inspiration and expiration ports are analyzed for gas radioactivity.

throughout the experiment and continually monitored for alertness or any other signs of anesthesia wearing off with pupillary reflex and toe-pinches. Inspired and expired gas concentrations were measured at 10 min intervals starting at 5 min, allowing the calculation of the inhaled dose as the difference between the inspired and expired concentrations. Following the 30 min experimental period the rat was euthanized by opening the heart and taking a blood sample whilst remaining under anesthesia. Euthanization was necessary as to obtain counts specific to each organ the organs needed to be removed, whilst any long-term complications due to radioactivity exposure was also ethically prevented. The brain, liver, femur (bone), and quadriceps (muscle) were then removed and weighed (CP64, Sartorius, Germany).

## Hyperbaric experiments

In hyperbaric experiments rats were investigated in pairs, with rats randomly placed into induction boxes, one of which received oxygen, and the other air, for 60 min (Fig 1). Upon arrival of the $^{13}N_2$ both rats were placed inside a plastic cage without food or water inside a hyperbaric chamber (50 L, OxyCure 3000 Hyperbaric Incubator, OxyHeal Health Group, CA, USA). After the addition of $^{13}N_2$ laden air, the chamber pressure was increased to 700 kPa (absolute) at a rate of ~240 kPa.min$^{-1}$. Rats could be continually monitored using a video monitor. The pressure was then held at 700 kPa for 30 minutes before decompression commenced at a rate of ~70 kPa.min$^{-1}$ until pressure reached 200 kPa, whereupon regular air was added to flush $^{13}N_2$ from the chamber for safety reasons, resulting in an increase in pressure back to 275 kPa (Fig 3). Pressure was then reduced to 200 kPa and the flush was repeated two more times before being reduced back to ambient pressure (100 kPa) with an approximate total dive time of 45 min.

Following removal from the chamber both rats were anaesthetized using isoflurane operated with air. The rats were then euthanized and had their organs removed in line with the normobaric trial.

## Organ counting

All organs were placed into a gamma counter (2480 WIZARD$^2$, PerkinElmer, Singapore) which measured gamma radiation within a 400–600 keV range (nitrogen-13 releases gamma rays at 511 keV) for 2 min for each sample. An empty vial was also measured to account for background radiation. All samples were corrected for radioactive decay. Samples were then normalized to organ mass. As blood $^{13}N_2$ content theoretically represents the $^{13}N_2$ delivered to organs samples were also normalized to a proportion of blood content.

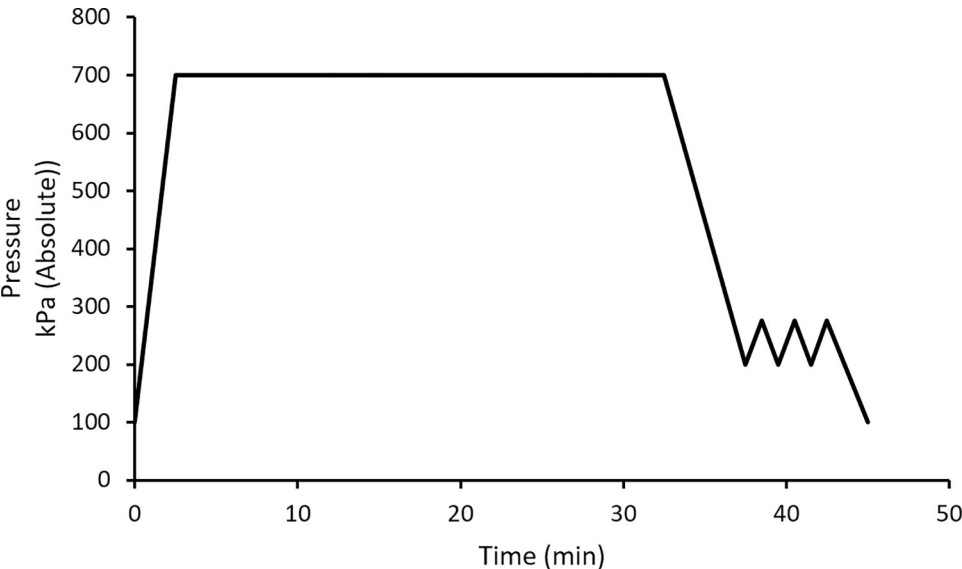

**Fig 3. Dive profile of the rats involving compression to 700 kPa, before a longer decompression period that included flushing the chamber to minimize safety risks due to using radioactive $^{13}N_2$.**

All radioactivity measurements (including those of the $^{13}N_2$ vial) were corrected for radioactive decay using the following formula:

$$A_0 = \frac{A(t)}{e^{-\lambda \cdot t}} \qquad \text{Eq1}$$

Where $A_0$ is baseline counts per minute, $A(t)$ is counts per minute at the time point ($t$), and $\lambda$ is equal to 0.693/9.965 min, where 0.693 is equal to the natural logarithm of 2, and 9.965 min is the half-life of $^{13}N_2$.

## Statistical analysis

Main outcome variables were tested for normality using Q-Q plots and Shapiro-Wilk tests, and were found to not be normally distributed. As such non-parametric tests were used throughout, which also supports the small sample sizes used. Main effect comparisons between groups were made using Kruskall-Wallis rank sum tests in R for Statistics (Version 4.2.1). All data are expressed as mean (± standard deviation). Standard error was calculated by dividing the standard deviation by the square root of the number of subjects.

## Results

All ten rats in the hyperbaria condition successfully completed the experiment. However, two rats in the normobaric conditions died during the procedure (1 air prebreathe, 1 OPB), and an issue with the gamma counter resulted in an OPB experiment returning no results. This resulted in the normobaric air prebreathe having 4 subjects, while the normobaric OPB had only 3 subjects.

## $^{13}N_2$ doses

Normobaric rats prebreathed oxygen for 64.8 (± 7.1) min, whereas hyperbaric rats prebreathed oxygen for 51.9 (± 11.2) min before the experiments began. In the normobaric experiments

**Table 1.** Absolute blood counts per minute following exposure to either normobaria (100 kPa) or hyperbaria (and subsequent decompression; 700 kPa) after prebreathing either air or oxygen for 60 min.

|  | Mean | SD |
|---|---|---|
| Normobaria, oxygen prebreathe | 5182.2 | 1207.6 |
| Normobaria, air prebreathe | 1887.9 | 1341.6 |
| Hyperbaria, oxygen prebreathe | 73.1 | 12.3 |
| Hyperbaria, air prebreathe | 89.3 | 30.7 |

the OPB saw 90.5 (± 25.2) Mbq (Megabecquerel–radioactivity unit) extracted into the bag, of which 22.6 (± 16.5) Mbq was taken up during the experiment. In the air prebreathe 103.4 (± 17.4) Mbq was extracted, similar to that in the OPB (p = .724), as was the uptake of 9.9 (± 6.1) Mbq (p = .724). In the hyperbaric experiments a mean dose of 140.82 (± 54.59) Mbq was extracted into the chamber. Compression lasted 2.7 (± 0.3) min, before 45 (± 0) min at 700 kPa, followed by a decompression of 12.3 (± 1.1) min.

## Absolute blood counts

Absolute blood counts were much lower in the hyperbaric conditions than in normobaria (p = .001; Table 1). Within the normobaria conditions the oxygen prebreathe had larger counts than the air prebreathe (p = .034), however there was no effect of prebreathe in the hyperbaric conditions (p = .465).

## Organ counts

Within normobaria a similar effect of prebreathe was seen in the uptake in all organs when normalized to blood value (p = .034). Specifically, the liver and muscle had significantly higher counts (all p = .034; Fig 4A), whereas the difference in the brain (p = .077) and bone (p = .157) did not reach significance. However, these effects were not observed in the hyperbaric conditions for any organ (all p>.344; Fig 4B). Pooled comparisons between hyperbaria and normobaria showed both bone and muscle normalized counts were higher in hyperbaria (p = .015; Fig 4), with no difference in brain (p = .204) or liver (p = .558).

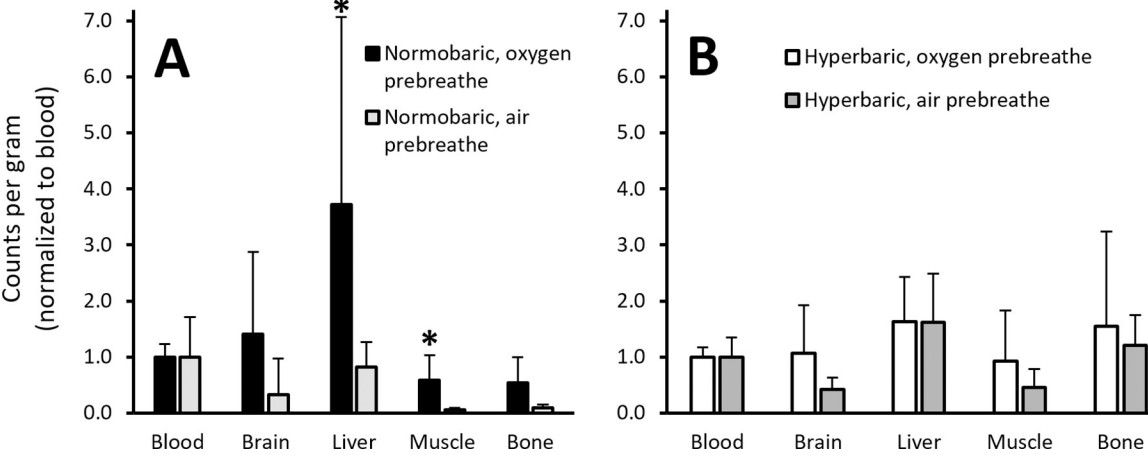

**Fig 4.** Nitrogen-13 counts, measured by a gamma counter, in organs normalized to mass and blood values in rats exposed to normobaria (Panel A; 100 kPa) or hyperbaria (Panel B; 700 kPa) after prebreathing either air or oxygen (60 min). *p < .05 between prebreathe conditions within pressure conditions.

## Discussion

The differences observed in normobaria, with more $^{13}N_2$ present in oxygen-prebreathed (OPB) tissues, suggests that the prebreathe procedure allowed for the off-gassing of nitrogen, effectively creating a "sink" for $^{13}N_2$ to move into. However, under hyperbaric conditions the effects of the prebreathe were not observed.

### Normobaric environment

In normobaria, the OPB resulted in an uptake of nearly three times as much $^{13}N_2$ into blood (Table 1). In theory, the amount of nitrogen present in the blood of both rats would be identical, with the difference being that naturally existing nitrogen-14 is replaced with $^{13}N_2$. Therefore, the OPB removed ~64% of nitrogen from the blood. This supports the practice of using OPB to off-gas nitrogen to prevent decompression sickness in aviators undergoing fast, unpressurized ascent, and astronauts doing extra-vehicular activities [4, 9, 10]. Both liver and muscle were also strongly affected by OPB (Fig 4A). This suggests that these areas were the easiest to consistently remove nitrogen from during the OPB. Muscle counts were nearly 10x greater following OPB, suggesting that the risk of muscular DCS on upon decompression, if breathing a nitrogen-less gas, would be 10x less. Indeed, all measured organs in the normobaric condition saw clinically meaningful benefits of OPB, with at least 4-fold increases in $^{13}N_2$ counts. As the normobaric data is likely low on statistical power due to the incomplete trials it is possible that further experiments would reveal statistically significant differences.

### Hyperbaric environment

The hyperbaric condition had lower absolute blood counts of $^{13}N_2$ (Table 1). There are two main reasons for this observation. The first is the difference in the time between the end of $^{13}N_2$ breathing and dissection. In the hyperbaric experiments the time taken for decompression and sedation (~15 min) allows the tissues to release $^{13}N_2$ from tissues down a partial pressure nitrogen gradient, whereas there was minimal time for any off-gassing in the normobaric trials as euthanasia happened immediately upon cessation of $^{13}N_2$ breathing. Secondly, both the chamber volume and the additional air required to induce hyperbaria, dilutes the amount of $^{13}N_2$ in each breath, therefore leading to reduced absolute $^{13}N_2$ uptake. Conversely, in the normobaric experiments rats were intubated and were being directly supplied with the $^{13}N_2$ dose. There was also no difference in $^{13}N_2$ content in any organ between prebreathe groups observed in hyperbaria following OPB (Table 1). The large pressure exposure likely minimizes any effects of OPB. The pressure gradient moving nitrogen into tissues is 7x greater in the hyperbaric condition. As this essentially increases the amount of nitrogen that can move into each tissue [11], it off-sets the larger capacity for nitrogen uptake seen in the OPB at normobaria. Theory suggests a slight difference should exist, and maybe does at lower hyperbaric pressures. It is unclear if under the employed hyperbaric pressure (700 kPa) such an effect is beyond the within-subjects variation [12], and could be measured with the limitations of the current experimental set-up. The main limitation is the requirement of decompression, that allows off-gassing for ~15 min. This dilutes any difference in tissue nitrogenation that may exist. Indeed differences have been shown with OPB prior to diving, where bubble formation is reduced [13]. However, this involved a similar profiled dive to 400 kPa. Based on the results in this study such an effect would be stronger at lower pressure exposures, so whether a detectable finding in humans would be observed at 700 kPa is unknown.

Despite no differences in $^{13}N_2$ counts in any organ due to OPB, when both hyperbaric conditions were pooled there was an overall effect in muscle and bone compared to the normobaric groups (Fig 4). It is likely that these tissues take longer to take up, and release, nitrogen, and

therefore are known as "slow tissues" [14]. While this is likely due to factors such as lower relative blood flow (especially as minimal musculoskeletal activity was occurring; [15, 16]), other factors pertaining to the blood-tissue interface remain undetermined [17], largely due to a lack of ability to measure nitrogen movement across this interface. The result of these slower tissues is that it takes them longer to off-gas the nitrogen compared to faster tissues such as brain and liver [18]. As such, following the ~15 min off-gassing period (largely caused by decompression) these faster tissues can release their $^{13}N_2$ back into the bloodstream and subsequently expire it, whereas the slower tissues release less, and therefore hold onto more $^{13}N_2$. To observe how much each organ was affected at depth could be determined by euthanizing the rat at depth using carbon dioxide, to firstly prevent further respiration that removes $^{13}N_2$ from the system, and secondly to allow rapid decompression without causing pain to the experimental animal.

## Conclusions

This study presents a method that can quantify gas uptake in normobaric and hyperbaric conditions within individual tissues. In both hyperbaric and normobaric experiments $^{13}N_2$ uptake aligned with our current understanding of nitrogen kinetics in decompression sickness.

## Acknowledgments

We would like to acknowledge Chris Barback for his assistance in setting up the lab for these experiments.

## Author Contributions

**Conceptualization:** Peter Lindholm.

**Data curation:** Edward Tom Ashworth, Ryotaro Ogawa.

**Formal analysis:** Edward Tom Ashworth, David Robert Vera.

**Funding acquisition:** Peter Lindholm.

**Investigation:** Edward Tom Ashworth, Ryotaro Ogawa.

**Methodology:** Edward Tom Ashworth, Ryotaro Ogawa, Peter Lindholm.

**Project administration:** Edward Tom Ashworth, Ryotaro Ogawa, David Robert Vera.

**Resources:** Edward Tom Ashworth, Ryotaro Ogawa, David Robert Vera.

**Software:** Edward Tom Ashworth.

**Supervision:** David Robert Vera, Peter Lindholm.

**Validation:** David Robert Vera, Peter Lindholm.

**Writing – original draft:** Edward Tom Ashworth.

**Writing – review & editing:** Edward Tom Ashworth, Ryotaro Ogawa, David Robert Vera, Peter Lindholm.

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
