## [Decision Letter · Decision Letter 0]

21 Aug 2023

PONE-D-23-17850Effects of Oxygen-Prebreathing on Tissue Nitrogenation in Normobaric and Hyperbaric ConditionsPLOS ONE

Dear Dr. Ashworth,

Thank you for submitting your manuscript to PLOS ONE. After careful consideration, we feel that it has merit but does not fully meet PLOS ONE’s publication criteria as it currently stands. Therefore, we invite you to submit a revised version of the manuscript that addresses the points raised during the review process.

We look forward to receiving your revised manuscript.

Kind regards,

Peter R. Corridon

Academic Editor

PLOS ONE

Reviewers' comments:

Reviewer's Responses to Questions

**Comments to the Author**

1. Is the manuscript technically sound, and do the data support the conclusions?

Reviewer #1: Partly

Reviewer #2: Yes

2. Has the statistical analysis been performed appropriately and rigorously? 

Reviewer #1: Yes

Reviewer #2: I Don't Know

3. Have the authors made all data underlying the findings in their manuscript fully available?

Reviewer #1: Yes

Reviewer #2: Yes

4. Is the manuscript presented in an intelligible fashion and written in standard English?

Reviewer #1: Yes

Reviewer #2: Yes

5. Review Comments to the Author

Reviewer #1: I am enthused about this work, and it does set a basis for a dramatic increased understanding of gas flow kinetics with changes in ambient pressure.

The methods are novel and difficult to accomplish.

Comments:

Introduction

This paragraph should be stronger. There are many forms of diving that are not SCUBA (self-contained breathing apparatus) used in the military and commercial (ie surface supplied and rebreather circuits). Probably the most at risk for problems are the civilian "technical divers".

Suggest something broader that will encompass both hyper and hypobaric decompression sickness. Something discussing inert gas being based on partial pressure and a decrease in ambient pressure leading to supersaturation and bubble formation etc. I think a DCS introductory paragraph is generally difficult.

Would add second paragraph on basics of decompression, with allowing the safe exhalation of excess inert gas.

Line 44: There is Oxygen pre-breath (OPB) data in hyperbaric environments. The Navy Experimental Diving Unit (NEDU) has studied this in humans pertaining to disabled submarine rescue (Latson 2000). This form of OPB is prior to decompression. Reference 12 can be used in this paragraph which mimics the type of OPB studied

Line 55: Consider using OPB (oxygen pre-breathe)

Line 64-69: the OPB methods are best placed in their respective sections.

Figure 1: Explanation is incomplete.

Line 77: Is the font for "PhysioSuite" off (perhaps my vision)?

Line 80-83. OPB is not mentioned. Perhaps comment on randomization(yes/no)

Line 89-91: Seems the euthanasia aspect does not need a justification line, but defer to authors.

FIGURE 2: This is not the experiment reported. This seems to be figure from Reference 7 where a PET scanner was used and not a gamma counter.

Lines 98-106: OPB not mentioned here. Again, were they randomized? After flushing with 13N2 was the chamber then pressure using regular air ? (this is stated in discussion BUT should be in methods primarily)

Line 142: Mbq should be defined.

Line 149: as you acknowledge in discussion, lower 13N2 levels in blood may NOT be indicative of lower 13N2 delivered and can be a function of elimination

Line 139-152: I simply cannot derive any meaning from the hyperbaric studies. The "dilution" of 13N2 with 140 Mbq into a 50L chamber then compression to 700kPa alone comes close to explaining the low serum 13N2. This is then followed by a significant decompression (1/3 the length of the dive) that included air flushes. In my opinion this is a study on OPB effects on nitrogen "loading" AND a decompression profile. For me, too many unknowns/confounders are at play to make any comments on positives or negatives of OPB meaningless.

Nonetheless using 13N2 in a hyperbaric environment has real potential.

The authors do well to describe the issues in discussion BUT conclusions are drawn based on problematic methods as they relate to the goals of evaluating N2 loading with OPB. In addition to lower chamber volume and lower kPa, animals can be euthanized prior to surfacing by breathing only inert gas.

Given the expense of 13N2, the complexity of working with the isotope, I suspect that this phase of the study is pilot work. As pilot work it is highly commendable. However, I would avoid drawing conclusions. I suggest including this phase as proof of concept and a demonstration that 13N2 can be a highly useful tool. Another option would be removing the hyperbaric studies from the manuscript BUT I think that is not very helpful.

With that in mind I would re-work all aspects related to the hyperbaric work including title, discussion, and conclusions.

I appreciate the opportunity to review this work. Again, the authors should be commended. Hopefully the comments are taken as constructive.

Reviewer #2: The authors have performed experiments in Sprague Dawley rats in order to examine effects of pre-breathe conditions on nitrogen washout from blood, brain, liver, femur and thigh muscle. There were two groups, breathing air with 13N2 for 30 minutes at either 1 or 7 ATA. Each group had two subgroups, that pre-breathed either air or 100% O2 while anesthetized with isoflurane. The hyperbaric group were decompressed and then euthanized using isoflurane, after which tissues were obtained. In the normal Barrick exposed group euthanize Asian and dissection occurred immediately at the end of the 30 minute exposure. 13N2 activity was assessed using a gamma counter. Liver and muscle had higher counts after the oxygen pre-breathe, with generally higher counts in both groups that had been exposed to hyperbaric air.

The results are not surprising. As expected, hyperbaric nitrogen exposure resulted in higher nitrogen levels. Similarly, those animals that pre-breathed oxygen had higher nitrogen levels, as explained by the authors due to the increased quote sink unquote for nitrogen.

The experiment did not provide any new insight into nitrogen uptake or washout. Since the introduction contained some discussion about oxygen pre-breathe prior to altitude exposure, it might have been more interesting to look at nitrogen washout during hypobaric exposure. Another approach might have assessed nitrogen washout during either normobaric or hyperbaric oxygen breathing, when peripheral vasoconstriction might delay inert gas washout.

However, the experiment is of huge interest due to what I believe is a new technique. To my knowledge, this is the first attempt to use 13N2 as an inert gas marker for gas tissue washout experiments related to decompression sickness. A particularly useful way to improve the paper would be to outline this technique in more detail, with discussion about experimental error as well as other possible applications of the technique. Given the short half-life of 13N2, how precise must the timing of measurements be? How long could an experiment last before the radioactivity had decayed to an unquantifiable level? Would it be possible to image animals after decompression using positron emission tomography? In this way the paper could be made much more appealing by editing the manuscript into a methods paper and using a somewhat different title, similar to their abstract at the American Physiology Summit. Their experimental results would then be more appropriate as an example of how to use the technique, rather than the main focus.

Minor point:

In Figure 1, add the isoflurane administration to the timeline.

6. PLOS authors have the option to publish the peer review history of their article (what does this mean?). If published, this will include your full peer review and any attached files.

Reviewer #1: No

Reviewer #2: **Yes: **Richard Moon

---

## [Author Response · Author response to Decision Letter 0]

22 Sep 2023

We would like to extend our thanks to both reviewers for their feedback on this manuscript. We believe the changes that we have made in response to this feedback have strengthened the work from a scientific perspective. Please find responses to your comments available at the very bottom of the PDF with responses in red, and the attached ‘tracked-changes’ version of the document, to which any references to line numbers in our responses refer, just above that. A clean copy of the manuscript is also available.

Reviewer #1: I am enthused about this work, and it does set a basis for a dramatic increased understanding of gas flow kinetics with changes in ambient pressure.

The methods are novel and difficult to accomplish.

Comments:

Introduction

This paragraph should be stronger. There are many forms of diving that are not SCUBA (self-contained breathing apparatus) used in the military and commercial (ie surface supplied and rebreather circuits). Probably the most at risk for problems are the civilian "technical divers".

Suggest something broader that will encompass both hyper and hypobaric decompression sickness. Something discussing inert gas being based on partial pressure and a decrease in ambient pressure leading to supersaturation and bubble formation etc. I think a DCS introductory paragraph is generally difficult.

Would add second paragraph on basics of decompression, with allowing the safe exhalation of excess inert gas.

The introduction has been slightly adapted to include diving beyond SCUBA, and include civilian divers alongside other hypo and hyperbaric DCS (line 36)

Line 44: There is Oxygen pre-breath (OPB) data in hyperbaric environments. The Navy Experimental Diving Unit (NEDU) has studied this in humans pertaining to disabled submarine rescue (Latson 2000). This form of OPB is prior to decompression. Reference 12 can be used in this paragraph which mimics the type of OPB studied

Thank you for bringing this study to our attention. We have added this to the introduction as suggested (line 46)

Line 55: Consider using OPB (oxygen pre-breathe)

Thank you for the suggestion we have implemented this (line 45 onwards)

Line 64-69: the OPB methods are best placed in their respective sections.

This part of the method has been reduced and placed in the normobaric/hyperbaric experiment sections (line 68-70, 107-108)

Figure 1: Explanation is incomplete.

Figure caption has now been completed (line 76-77)

Line 77: Is the font for "PhysioSuite" off (perhaps my vision)?

Thanks for noticing this, the font size has been corrected (line 82)

Line 80-83. OPB is not mentioned. Perhaps comment on randomization(yes/no)

The prebreathe has now been mentioned by moving it as suggested above. Randomization has been elaborated on (line 88, 107)

Line 89-91: Seems the euthanasia aspect does not need a justification line, but defer to authors.

This aspect is included per journal regulations

FIGURE 2: This is not the experiment reported. This seems to be figure from Reference 7 where a PET scanner was used and not a gamma counter.

The figure has been updated to reflect the current experiment (line 102)

Lines 98-106: OPB not mentioned here. Again, were they randomized? After flushing with 13N2 was the chamber then pressure using regular air ? (this is stated in discussion BUT should be in methods primarily)

Prebreathe information included here as done for the normobaric experiments. Detail added regarding the flushing with regular air (line 113)

Line 142: Mbq should be defined.

Added definition in text (line 152)

Line 149: as you acknowledge in discussion, lower 13N2 levels in blood may NOT be indicative of lower 13N2 delivered and can be a function of elimination

You are correct, and as such we have removed this statement (line 160)

Line 139-152: I simply cannot derive any meaning from the hyperbaric studies. The "dilution" of 13N2 with 140 Mbq into a 50L chamber then compression to 700kPa alone comes close to explaining the low serum 13N2. This is then followed by a significant decompression (1/3 the length of the dive) that included air flushes. In my opinion this is a study on OPB effects on nitrogen "loading" AND a decompression profile. For me, too many unknowns/confounders are at play to make any comments on positives or negatives of OPB meaningless.

We agree that there are major experimental differences between the normobaric and hyperbaric conditions that make comparisons between them vague at best. This has not been our intention, but rather to demonstrate the method (and its outcomes) at normobaria, and then in hyperbaria, where the method is of more practical use. The low serum 13N2 in the hyperbaric conditions is only low when directly compared to the normobaric experiment. To this end we have divided the discussion into a normobaric and hyperbaric section to minimize comparisons between the two experiments, and split the graph into 2 parts to further prevent comparisons (line 172). 

Once we could do the experiment at normobaria the next step was determining whether we could replicate an actual dive at hyperbaria to show the physiological process using our new method. This involves the compression and decompression periods that would be experienced by a diver, followed by a period of breathing normobaric air. We chose a single depth at the limit of the hyperbaric stimulus we could provide to provide a full view of the range of possibilities. We could have assessed other depths or time-points, but did not because we decided that we would work on improving the chamber design to maximize the 13N2 loading, and then address this in future studies. It is possible that a shorter dive would produce an effect (and a statement to this effect has been included – line 218-219), and indeed the basis of dive tables comes from similarly running iteration of experiments to determine the time-points at which this happens. Again, these are things we hope to investigate in the future once we address some of the limitations in the hyperbaric chamber that we discuss. 

Nonetheless using 13N2 in a hyperbaric environment has real potential.

The authors do well to describe the issues in discussion BUT conclusions are drawn based on problematic methods as they relate to the goals of evaluating N2 loading with OPB. In addition to lower chamber volume and lower kPa, animals can be euthanized prior to surfacing by breathing only inert gas.

Given the expense of 13N2, the complexity of working with the isotope, I suspect that this phase of the study is pilot work. As pilot work it is highly commendable. However, I would avoid drawing conclusions. I suggest including this phase as proof of concept and a demonstration that 13N2 can be a highly useful tool. Another option would be removing the hyperbaric studies from the manuscript BUT I think that is not very helpful.

Thank you for these comments. The conclusions have been adapted to reflect this exploratory nature of the research (line 243-244)

With that in mind I would re-work all aspects related to the hyperbaric work including title, discussion, and conclusions.

I appreciate the opportunity to review this work. Again, the authors should be commended. Hopefully the comments are taken as constructive.

Reviewer #2: The authors have performed experiments in Sprague Dawley rats in order to examine effects of pre-breathe conditions on nitrogen washout from blood, brain, liver, femur and thigh muscle. There were two groups, breathing air with 13N2 for 30 minutes at either 1 or 7 ATA. Each group had two subgroups, that pre-breathed either air or 100% O2 while anesthetized with isoflurane. The hyperbaric group were decompressed and then euthanized using isoflurane, after which tissues were obtained. In the normobaric exposed group euthanasia and dissection occurred immediately at the end of the 30 minute exposure. 13N2 activity was assessed using a gamma counter. Liver and muscle had higher counts after the oxygen pre-breathe, with generally higher counts in both groups that had been exposed to hyperbaric air.

The results are not surprising. As expected, hyperbaric nitrogen exposure resulted in higher nitrogen levels. Similarly, those animals that pre-breathed oxygen had higher nitrogen levels, as explained by the authors due to the increased quote sink unquote for nitrogen.

The experiment did not provide any new insight into nitrogen uptake or washout. Since the introduction contained some discussion about oxygen pre-breathe prior to altitude exposure, it might have been more interesting to look at nitrogen washout during hypobaric exposure. Another approach might have assessed nitrogen washout during either normobaric or hyperbaric oxygen breathing, when peripheral vasoconstriction might delay inert gas washout.

We anticipate the experiments detailed in the current manuscript being the beginning of a range of experiments that will explore these avenues you suggest along with others. While we do not currently have access to a controlled hypobaric chamber, we believe that the probabilistic nature of DCS linked to nitrogen load enables us to make the assessment that normobaric oxygen prebreathing does reduce the risk of DCS before altitude. As you mention, this is not a new insight, but merely a method showing, and quantifying, it in a way that hasn’t been done before. 

However, the experiment is of huge interest due to what I believe is a new technique. To my knowledge, this is the first attempt to use 13N2 as an inert gas marker for gas tissue washout experiments related to decompression sickness. A particularly useful way to improve the paper would be to outline this technique in more detail, with discussion about experimental error as well as other possible applications of the technique. Given the short half-life of 13N2, how precise must the timing of measurements be? How long could an experiment last before the radioactivity had decayed to an unquantifiable level? Would it be possible to image animals after decompression using positron emission tomography? In this way the paper could be made much more appealing by editing the manuscript into a methods paper and using a somewhat different title, similar to their abstract at the American Physiology Summit. Their experimental results would then be more appropriate as an example of how to use the technique, rather than the main focus.

We agree that there are details of the new technique that have not been included in this manuscript. The main reason for this is we are currently publishing the method as a standalone paper in a more methods-specific journal, that includes how we make the gas, the radio-physics that accompanies the method, and limitations and possible experimental in further detail. To ensure this data was available for more applied studies, such as the current manuscript, we published the pre-print of this method. We have added additional references to this text in areas that need it. The ability to correct for radioactive decay means that the timing of measurements merely needs to be recorded (± 1 min) to allow adequate correction using the Eq. 1. The length of experiments is dictated by the radioisotope half-life (10 min) allowing ~100 min of experimental time, and this has been included in the document (line 85). 

Minor point:

In Figure 1, add the isoflurane administration to the timeline.

Thanks for this suggestion, we have added this as suggested (line 76)

---

## [Decision Letter · Decision Letter 1]

6 Nov 2023

Effects of Oxygen-Prebreathing on Tissue Nitrogenation in Normobaric and Hyperbaric Conditions

PONE-D-23-17850R1

Dear Dr. Ashworth,

We’re pleased to inform you that your manuscript has been judged scientifically suitable for publication and will be formally accepted for publication once it meets all outstanding technical requirements.

Kind regards,

Peter R. Corridon

Academic Editor

PLOS ONE

Additional Editor Comments (optional):

Reviewers' comments:

Reviewer's Responses to Questions

**Comments to the Author**

1. If the authors have adequately addressed your comments raised in a previous round of review and you feel that this manuscript is now acceptable for publication, you may indicate that here to bypass the “Comments to the Author” section, enter your conflict of interest statement in the “Confidential to Editor” section, and submit your "Accept" recommendation.

Reviewer #1: All comments have been addressed

Reviewer #2: All comments have been addressed

2. Is the manuscript technically sound, and do the data support the conclusions?

Reviewer #1: Yes

Reviewer #2: Yes

3. Has the statistical analysis been performed appropriately and rigorously? 

Reviewer #1: Yes

Reviewer #2: Yes

4. Have the authors made all data underlying the findings in their manuscript fully available?

Reviewer #1: Yes

Reviewer #2: Yes

5. Is the manuscript presented in an intelligible fashion and written in standard English?

Reviewer #1: Yes

Reviewer #2: Yes

6. Review Comments to the Author

Reviewer #1: Thank you for your responsiveness to suggestions.

I have no further suggestions/comments

I agree that it is an improved manuscript and look forward to reading the published version.

Best wishes with furthering the technique/experiments

Reviewer #2: My concerns have been adequately addressed. This is a fabulous technique and I look forward to future investigations using it.

7. PLOS authors have the option to publish the peer review history of their article (what does this mean?). If published, this will include your full peer review and any attached files.

Reviewer #1: **Yes: **Richard T. Mahon

Reviewer #2: **Yes: **Richard Moon

---

## [Editor Report · Acceptance letter]

12 Jan 2024

PONE-D-23-17850R1 

PLOS ONE

Dear Dr. Lindholm, 

I'm pleased to inform you that your manuscript has been deemed suitable for publication in PLOS ONE. Congratulations! Your manuscript is now being handed over to our production team.

Kind regards, 

on behalf of

Dr. Peter R. Corridon 

Academic Editor

PLOS ONE